# Exergy Analysis for Utilizing Latent Energy of Thermal Energy Storage System in District Heating

**Joong Yong Yi [1], Kyung Min Kim [2],\***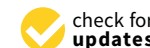**, Jongjun Lee [2] and Mun Sei Oh [2]**

1    Yujin Energy Consulting Co., Gyeonggi-do 14047, Korea; swsupport@naver.com
2    Korea District Heating Corp., Gyeonggi-do 17099, Korea; leejj@kdhc.co.kr (J.L.); sunny@kdhc.co.kr (M.S.O.)
\*    Correspondence: kimkm@kdhc.co.kr; Tel.: +82-31-8014-9645

**Abstract:** The thermal energy storage (TES) system stores the district heating (DH) water when the heating load is low. Since a TES system stores heat at atmospheric pressure, the DH water temperature of 115 °C has to be lowered to less than 100 °C. Therefore, the temperature drop of the DH water results in thermal loss during storage. In addition, the DH water must have high pressure to supply heat to DH users a long distance from the CHP plant. If heat is to be stored in the TES system, a pressure drop in the throttling valve occurs. These exergy losses, which occur in the thermal storage process of the general TES system, can be analyzed by exergy analysis to identify the location, cause and the amount of loss. This study evaluated the efficiency improvement of a TES system through exergy calculation in the heat storage process. The method involves power generation technology using the organic Rankine cycle (ORC) and a hydraulic turbine. As a result, the 930 kW capacity ORC and the 270 kW capacity hydraulic turbine were considered suitable for a heat storage system that stores 3000 $m^3$/h. In this case, each power generation facility was 50% of the thermal storage capacity, which was attributed to the variation of actual heat storage from the annual operating pattern analysis. Therefore, it was possible to produce 1200 kW of power by recovering the exergy losses. The payback period of the ORC and the hydraulic turbine will be 3.5 and 7.13 years, respectively.

**Keywords:** district heating; latent energy; power generation; thermal energy storage

## 1. Introduction

A recent projection predicted that primary energy consumption will rise by 48% in 2040 [1]. The negative environmental impact of fossil resources has accelerated the shift toward renewable energy sources. In the development of renewable energy systems, energy storage has become an important part of renewable energy technology. Thermal energy storage (TES) is a technology that stocks thermal energy by heating or cooling so that the stored energy can be used for heating and cooling applications [2] and power generations at later time. TES systems are used particularly in buildings and in industrial processes. Advantages of TES in energy systems include an increase in overall efficiency and better reliability. TES can also lead to better economics, reductions in investment and running costs, and less pollution of the environment, i.e., fewer carbon dioxide emissions [3]. TES is becoming particularly important for electricity storage in combination with concentrating thermal power plants where heat can be stored for electricity production when operation is not economic. In Europe, it has been estimated that around 1.4 million GWh per year can be saved and 400 million tons of $CO_2$ emissions avoided, in buildings and in industrial sectors, by more extensive use of heat and cold storage [4].

Storage density, in terms of the amount of energy per unit volume or mass, is important for optimizing the solar ratio (how much solar radiation is useful for the heating/cooling purposes), the

efficiency of the appliances (solar thermal collectors and absorption chillers), and energy consumption for space heating/cooling consumption. Phase-change materials (PCMs) in solar systems are worth investigating. PCMs might be able to increase the energy density of small-sized water storage tanks, reducing solar storage volume for a given solar fraction or increasing the solar fraction for a given available volume [5,6]. Other studies [7,8] have been performed using advanced thermodynamic analysis for thermal storage. They found the optimal values of PCMs.

While heat storage on the hot side of thermal plants is always present because of heating and/or domestic hot water (DHW) production, cold storage is justified in larger plants. Cold storages are used not only for the economic advantages of lower electricity costs (in the case of electric compression chillers), depending on the time of day, but also to lower the installed cooling power and to allow more continuous operation of the chiller. The use of thermal storage could not provide effective backup but helped the system thermally stabilize. Since then, studying thermal energy storage technologies, as well as the usability and effects of both sensible and latent heat storage in numerous applications, has increased, leading to a number of reviews [9]. These reviews focused only on one side (cold or hot), a component of the system or one of its integral mechanisms.

District heating and cooling (DHC) with thermal energy storage (TES) can enhance energy efficiency. DHC with TES can also help address global environmental problems. DHC systems are required to maintain comfortable thermal conditions for the occupants of commercial and office buildings. Traditional heating, ventilating, air-conditioning and cooling (HVAC) systems are high costs and contribute to peak demand [10]. Thus, much research has been undertaken to reduce greenhouse gas (GHG) emissions using renewable energy and energy resources intelligently. DHC systems can use renewable energy and waste heat as a thermal energy source, and facilitate the intelligent integration of energy systems. The district heating system is a heating system that generates heat from large-scale heat source facilities such as cogeneration plants and supplies heat to nearby customers. Generally, the thermal energy storage system is equipped with a heat source system. TES systems can be generated with district heating (DH) equipment installed for the efficient operation of the heat source equipment for the district heating water (primary heating water). TES tanks in DHC systems, which store heating/cooling energy in the off-peak period for use in the peak period, can be used to offset peak time energy demand [10–12].

Since the heat storage system stores heat at atmospheric pressure, the primary heating water produced from the heat source equipment is stored at less than 100 °C (95~98 °C). If the temperature of the primary heating water is higher than the storage temperature, heat loss occurs. The stored DH water of 98 °C and 65 °C coexist in the TES tank. When the heat is stored, the 65 °C DH water in the tank is sent to heat the source equipment and then raise the temperature to the primary heating temperature. It is then stored in the TES system or supplied to the DH pipes. In addition, the primary heating water produced by the heat source equipment must maintain high pressure (more than 10 bar) to supply heat to long distance district heating customers. However, because the heat is stored at atmospheric pressure in the TES system, the pressure loss is generated in the heat-storage process. Therefore, there are two energy losses, heat loss and pressure loss, in the heat-storage process of a district heating system.

Exergy analysis deals with the quality of the energy based on the second law of thermodynamics. It is useful to identify the inefficiencies, their reasons, locations, and the amount of energy lost in the process of the energy system [13]. As an analytical method for comparing and evaluating TES systems, the exergy analysis method can be a very reasonable and meaningful alternative to the energy analysis method. Exergy analysis provides efficiency, which means determining the actual value of how the performance of the system approaches the ideal values by determining the amount of thermodynamic loss, the cause and the location [14]. Therefore, thermodynamic losses in the heat-storage process of the TES system are analyzable through exergy analysis. It is also possible to get substantial information and to obtain an optimum design for the improvement of the efficiency of the TES system.

A number of studies related to district heating systems have been undertaken. Rezaie et al. [15,16] analyzed the energy and exergy efficiency of the TES system in the district heating system. In the case of a district heating system including a solar panel, they explained that the TES system improves the effectiveness of the solar panel. Rosen and Dincer [17] emphasized two important factors, the proper efficiency analysis method and the proper temperature in evaluation of heat storage systems, as results of the exergy analysis of closed TES systems. They have shown that the exergy analysis method can be usefully applied to the optimization of operation and the optimal design of the TES system.

Recently, many studies have been carried out to cope with climate change and to reduce greenhouse gas emissions. In the field of district heating, many researchers are trying to reduce greenhouse gas emissions via improvement of the efficiency, application of renewable energy and so on. Robins [18] studied a variety of methods that could be practically applied to reduce GHG emissions. Rosen et al. [19–21] proposed alternative energy sources that can replace fossil fuels. Dincer [22] proposed environmentally friendly methods such as energy conservation, the application of renewable energy and clean energy technology.

Above all, the way to increase the utilization efficiency of energy sources is to use high-efficiency facilities or to recover lost or abandoned energy. Firstly, it is necessary to analyze the energy losses in temperature, pressure, electricity, or any other form, in a system, and then find ways to improve efficiency through energy recovery.

The previous studies have studied new TES system designs, new materials, and solar thermal storage systems. At present, however, many TES systems for storing hot water have been used at atmospheric pressure in industrial fields. This study investigated the energy loss that occurs in the process of storing heat in DH industries. A method for improving the efficiency of the TES system by utilizing the potential energy was studied. Using the exergy analysis method, the exergy values of the respective streams constituting the target system were calculated. The points where the exergy losses occurred and the amount of loss were also calculated in the TES system. In addition, the methods of reducing exergy losses in each process were proposed and additional generated energies were calculated.

## 2. Research Methods

### 2.1. Methods of Exergy Analysis

Exergy means the maximum available work that can be obtained theoretically when any system goes from a non-equilibrium to equilibrium state [23]. This can be expressed as:

$$E = E_{ph} + E_{kn} + E_{pt} + E_{ch}, \tag{1}$$

where $E_{kn}$ is exergy by velocity or kinetic energy, $E_{pt}$ is exergy by the potential energy, $E_{ph}$ is the physical exergy, which is the difference of temperature and pressure between actual state and reference state, and $E_{ch}$ is chemical exergy, resulting from the reaction. Physical exergy is expressed as follows:

$$E_{ph} = (U - U_0) + p_0(V - V_0) - T_0(S - S_0) = (h - h_0) - T_0(S - S_0) = W_{c,max}, \tag{2}$$

where the subscript "0" refers to the reference state as thermal, mechanical equilibrium for calculating the exergy value. In Equation (2) the physical exergy is the work obtained from the change of temperature and pressure in the system to the reference state.

The exergy components in the change process of the system are shown in Figure 1.

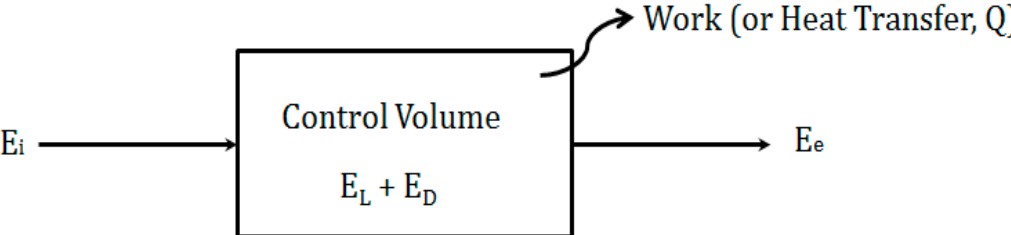

**Figure 1.** Exergy components in the change process.

The exergy balance for the change process of the system can be expressed by Equation (3). The inlet of exergy ($E_i$), which is the total exergy in the system, is expressed by the sum of exergy loss ($E_L$), the exergy destruction ($E_D$), and the outlet of exergy ($E_e$). Exergy loss ($E_L$) means loss of exergy, which does not contribute to work or heat transfer (Q). Exergy destruction ($E_D$) is exergy destroyed by friction within the control volume. They all have the same meaning in that they do not contribute to work or heat transfer in the process of changing the system [22].

$$E_i = E_e + E_L + E_D, \qquad (3)$$

Figure 2 is a schematic diagram showing a district heating system associated with a heat storage system. As shown in Figure 2, the thermal energy storage (TES) system includes a tank for storing heat generated from the heat source, a control valve for controlling the flow rate or pressure of the heating water during storage, and a release pump for supplying heat from the storage tank to the district heating customer. Here, a DH source (DH heater) for generating heat and a DH supply pump for supplying generated heat to the customer were added. In the study, the DH heater, release pump, and DH consumer were excluded from the analysis.

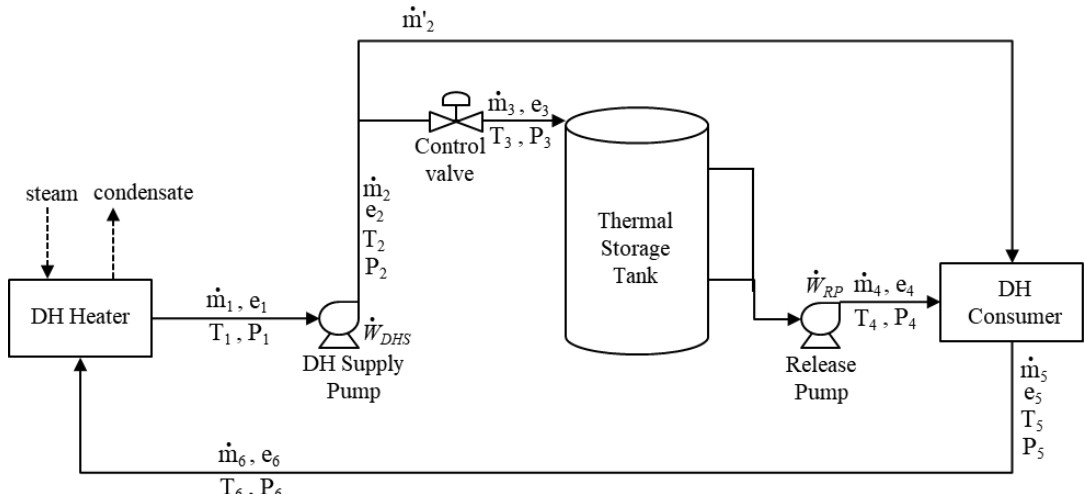

**Figure 2.** Schematic diagram of thermal energy system (TES) in district heating system.

For exergy analysis of the heat storage system, the exergy values for each stream should be calculated. Then, the exergy balance in the target process, based on the exergy values, should be calculated by Equations (2) and (3). In Equation (2), $T_0$ is the reference temperature for calculating the exergy value of each stream. As shown in Equation (2), the exergy value depends on the reference temperature and the result is the different exergy value. As a result, different exergy analysis results can be derived for the exergy value of each stream. Therefore, selection of the reference temperature in the exergy calculation is very important and should be done depending on the characteristics of the system to be analyzed. In this study, the temperature ($T_0$) and pressure ($P_0$) of the reference state for calculating the exergy value were set to the thermodynamic standard temperature of 25 °C.

In Figure 2, ṁ, T, and P denote the flow rate (kg/s), temperature (°C), and pressure (kg/cm$^2$g) of each stream and e is the exergy per unit mass (kJ/kg). Therefore, the exergy flow (kJ/s) per unit time of the $n_{th}$ stream in the target process can be obtained from the following equation:

$$E_n \text{ (kW)} = \dot{m}_n \text{ (kg/s)} \times e_n \text{ (kJ/kg)}. \tag{4}$$

The exergy balances, calculated by Equations (5)–(7), for each process in the thermal storage system are expressed below.

Equation (5) represents before/after the DH supply pump:

$$(\dot{m}_1 \cdot e_1) + \dot{W}_{DHS} = (\dot{m}_2 \cdot e_2) - E_{L,①} \text{ or } E_1 + \dot{W}_{DHS} = E_2 - E_{L,①}. \tag{5}$$

Equation (6) is representative of before/after the control valve:

$$(\dot{m}_3 - \dot{m}_2) \cdot e_2 = (\dot{m}_3 \cdot e_3) - E_{L,②} \text{ or } E_2 = E_3 - E_{L,②}. \tag{6}$$

Equation (7) is before/after the release pump:

$$(\dot{m}_3 \cdot e_3) + \dot{W}_{RP} = (\dot{m}_4 \cdot e_4) - E_{L,③} \text{ or } E_3 + \dot{W}_{RP} = E_4 - E_{L,③}. \tag{7}$$

### 2.2. Analysis of Supply and Return Temperature of District Heating Water

The district heating heat source equipment is a facility for generating the primary heating water, which supplies space heating and hot water to customers. The temperature of the primary heating water is changed by the outside temperature and heating load. As a result of the analysis of the temperature of the primary heating water, as shown in Table 1 and Figure 3, the highest temperature found was 119 °C but the lowest was 87 °C. During winter (October to April) season, the average supply temperature was 110 °C. Given that the district heating was mainly supplied in the winter season, it was reasonable to apply a supply temperature of 110 °C.

If the temperature of the primary heating water generated from the heat source equipment changed, the return temperature changed accordingly. In the past year, as shown in Table 2 and Figure 3, the highest temperature was 96 °C but the lowest temperature was 22 °C. During winter (October to April), it was reasonable to apply an average return temperature of 45 °C.

**Table 1.** Supply temperature (unit: °C).

| Division | Jan. | Feb. | Mar. | Apr. | May | Jun. | Jul. | Aug. | Sep. | Oct. | Nov. | Dec. |
|----------|------|------|------|------|-----|------|------|------|------|------|------|------|
| Highest | 119 | 118 | 117 | 117 | 110 | 110 | 109 | 109 | 112 | 110 | 114 | 116 |
| Average | 112 | 112 | 112 | 110 | 96 | 99 | 99 | 99 | 99 | 101 | 107 | 109 |
| Lowest | 105 | 104 | 104 | 92 | 87 | 89 | 92 | 89 | 88 | 94 | 97 | 100 |

**Table 2.** First heated return temperature (unit: °C).

| Division | Jan. | Feb. | Mar. | Apr. | May | Jun. | Jul. | Aug. | Sep. | Oct. | Nov. | Dec. |
|----------|------|------|------|------|-----|------|------|------|------|------|------|------|
| Highest | 50 | 49 | 51 | 54 | 96 | 91 | 91 | 65 | 61 | 62 | 67 | 51 |
| Average | 43 | 43 | 44 | 44 | 70 | 53 | 57 | 57 | 51 | 50 | 44 | 44 |
| Lowest | 37 | 38 | 36 | 35 | 33 | 38 | 48 | 45 | 22 | 39 | 38 | 39 |

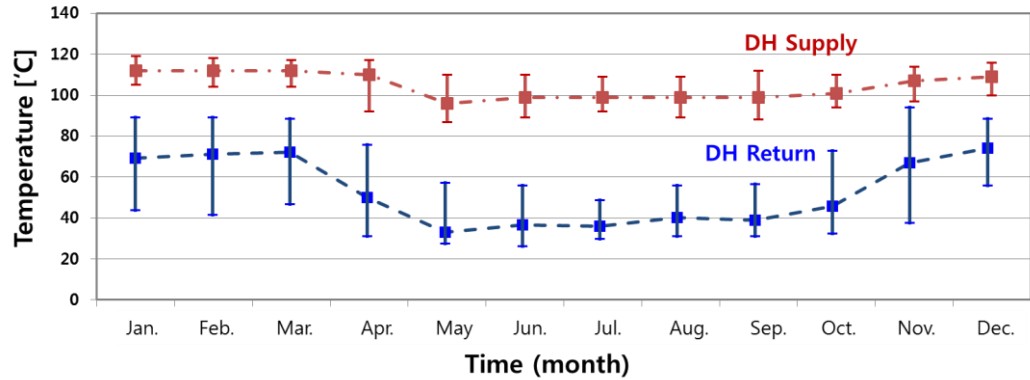

**Figure 3.** Temperature distributions of district heating (DH) supply and return water for a year.

## 2.3. Analysis of Supply Pressure During Storing Heat in TES

As mentioned, the primary heating water generated from the heat source equipment was stored in the TES system or supplied to customers depending on the heating load. At one point, the primary heating water was transported to long distance customers. Therefore, it should have been operated at a high pressure. The heat was stored in the TES system when the heat load was low but the pumping pressure was the same as when the heat was supplied to the long distance customers. Table 3 and Figure 4 show the analysis of the operational data of the supply pressure. The highest supply pressure of the primary heating water was 15.5 kg/cm$^2$g, the lowest was 4.3 kg/cm$^2$g, and the average was 11.6 kg/cm$^2$g.

**Table 3.** First heated supply pressure (unit: kg/cm$^2$g).

| Division | Jan. | Feb. | Mar. | Apr. | May | Jun. | Jul. | Aug. | Sep. | Oct. | Nov. | Dec. |
|---|---|---|---|---|---|---|---|---|---|---|---|---|
| Highest | 14.7 | 14.7 | 14.6 | 12.5 | 9.4 | 9.2 | 8.0 | 9.2 | 9.3 | 12.0 | 15.5 | 14.6 |
| Average | 11.4 | 11.7 | 11.9 | 8.2 | 5.4 | 6.0 | 5.9 | 6.6 | 6.4 | 7.5 | 11.0 | 12.2 |
| Lowest | 7.2 | 6.8 | 7.7 | 5.1 | 4.5 | 4.3 | 4.9 | 5.1 | 5.1 | 5.3 | 6.2 | 9.2 |

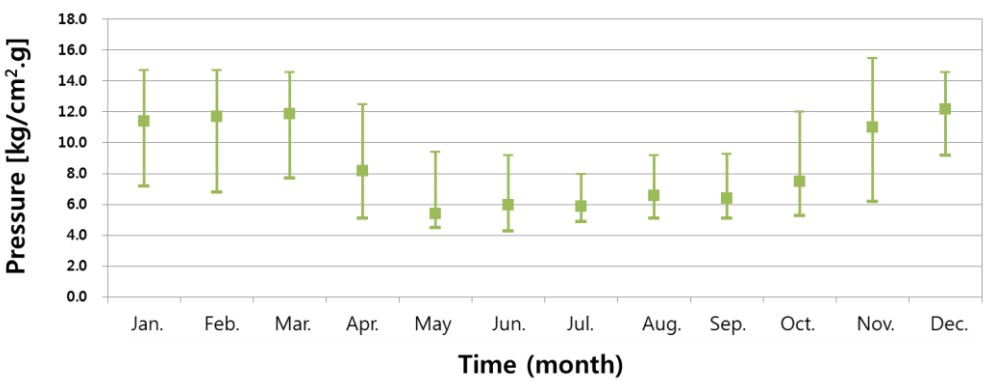

**Figure 4.** Pressure distributions of DH supply water for a year.

## 2.4. Analysis of Heat Storage Capacity

The storage tank in a TES system is designed based on a return temperature of 65 °C and a supply temperature of 98 °C (temperature difference of 33 °C) for 10 h of storage. In the case of a heat storage capacity of 1000 Gcal, the volume of storage is 30,000 m$^3$. It can pump a volume of 100 Gcal/h (volume of 3000 m$^3$/h) of DH water. Accordingly, when the heat storage tank capacity is known, the flow rate of the heating water from the amount of storing heat can be calculated. Table 4 presents the heat storage capacity designed with the operational data from the heat management system for TES system and a heat storage capacity of 1000 Gcal.

As a result, the maximum heat storage capacity in a year was 198 Gcal/h. In the winter, the average hourly capacity was about 50 Gcal/h. Considering that the heat capacity, per hour, of the storage tank is designed to be 100 Gcal/h, the average heat storage capacity was estimated to be half of the design capacity.

**Table 4.** Analysis of status of heat storage system operation (unit: Gcal/h).

| Division | Jan. | Feb. | Mar. | Apr. | May | Jun. | Jul. | Aug. | Sep. | Oct. | Nov. | Dec. |
|---|---|---|---|---|---|---|---|---|---|---|---|---|
| Highest | 135 | 153 | 120 | 165 | 156 | 138 | 145 | 172 | 169 | 158 | 198 | 145 |
| Average | 43 | 55 | 42 | 34 | 20 | 43 | 47 | 74 | 67 | 62 | 50 | 40 |
| Lowest | 14 | 16 | 14 | 12 | 6 | 15 | 14 | 20 | 20 | 19 | 17 | 14 |
| Time | 331 | 287 | 336 | 360 | 275 | 345 | 300 | 265 | 290 | 313 | 346 | 357 |

## 3. Results and Discussion

### 3.1. Exergy Analysis of the Previous System

Tables 5 and 6 show the results of the exergy calculation and exergy balance for the TES system based on the operational data analysis.

As a result of the exergy analysis for each process of the TES system, in all processes, ①, ②, and ③, exergy loss was found. The largest exergy loss appeared in Process ②. Respectively, the losses in Processes ① and ③ occurred in the pumps, the DH supply pump and release pump. These processes lost power by boosting the pressure necessary for supplying the primary heating water from the TES system and the heat source plant to the DH customer. However, the boost was an essential process for supplying heat to all of the customers, especially the long distance customers. There was a pressure loss and depending on the distance and the gradient. Therefore, it was reasonable to regard the exergy losses in Processes ① and ③ as exergy consumption not exergy loss.

In Process ②, storing DH water, the primary heating water was stored without work. In this process exergy loss occurred due to pressure loss by the throttling of the control valve and heat loss by temperature drop. The amount of exergy loss was approximately 26.6% of the exergy inlet. Therefore, it was necessary to find proper ways to reduce the exergy loss in the TES system to reduce the overall exergy loss.

**Table 5.** Exergy value calculation results for each flow of heat storage system.

| No | $\dot{m}$ (kg/s) | T (°C) | P (kg/cm$^2$g) | e (kJ/kg) | E (kW) | Remarks |
|---|---|---|---|---|---|---|
| 1 | 833.3 | 110 | 4.0 | 42.912 | 35,758.57 | $\dot{W}_{DHS}$ |
| 2 | 833.3 | 110 | 11.6 | 42.912 | 35,758.57 | =910.8 kW |
| 3 | 416.7 | 110 | 11.6 | 42.912 | 17,881.43 | =455.5 kW |
| 4 | 416.7 | 98 | 4.5 | 32.295 | 13,457.33 | $\dot{W}_{RP}$ |
| 5 | 416.7 | 98 | 11.6 | 32.295 | 13,457.33 | =425.5 kW |
| 6 | 833.3 | 44 | 4.0 | 2.427 | 2022.42 | |
| 7 | 833.3 | 44 | 4.0 | 2.427 | 2022.42 | |

**Table 6.** Estimation of exergy index for each process of the TES system.

| Process | Exergy Inlet (kW) | Exergy Outlet (kW) | Exergy Loss (kW) |
|---|---|---|---|
| ① DH Pump | 36,669.37 | 35,758.57 | $E_{L,①}$ = 910.8 |
| ② Control Valve | 18,336.93 | 13,457.33 | $E_{L,②}$ = 4879.6 |
| ③ Release Pump | 13,882.83 | 13,457.33 | $E_{L,③}$ = 425.5 |

As explained in this section, thermal loss and pressure loss were the causes of exergy loss in Process ②. The two types of exergy loss were the most common types of loss in the thermal systems. However, there were many ways to recover the loss. In this study, an organic Rankine cycle to reduce

heat loss and a hydraulic turbine to reduce pressure loss were applied. Exergy recovery was analyzed by the application of organic Rankine cycles and hydraulic turbines.

*3.2. Reduction of Exergy Losses from Thermal Loss*

An organic Rankine cycle was applied to reduce exergy loss from temperature loss. The organic Rankine cycle uses organic material (hydrocarbon compound) as the working fluid, not water, and consists largely of the evaporator, condenser, pump and heat exchanger. Applying an organic Rankine cycle to reduce exergy loss can produce power through recovery of exergy loss. The organic Rankine cycle system can be introduced as shown in Figure 5, based on the analysis of the operating state of the thermal storage system. The boundary conditions of the organic Rankine cycle are as follows:

- Amount of heat source water (primary heating water): 1500 m$^3$;
- Temperature of heat source water: 110 °C;
- Temperature of condensing water (return DH water): 45 °C;
- Working fluid: R245fa (environment-friendly refrigerant).

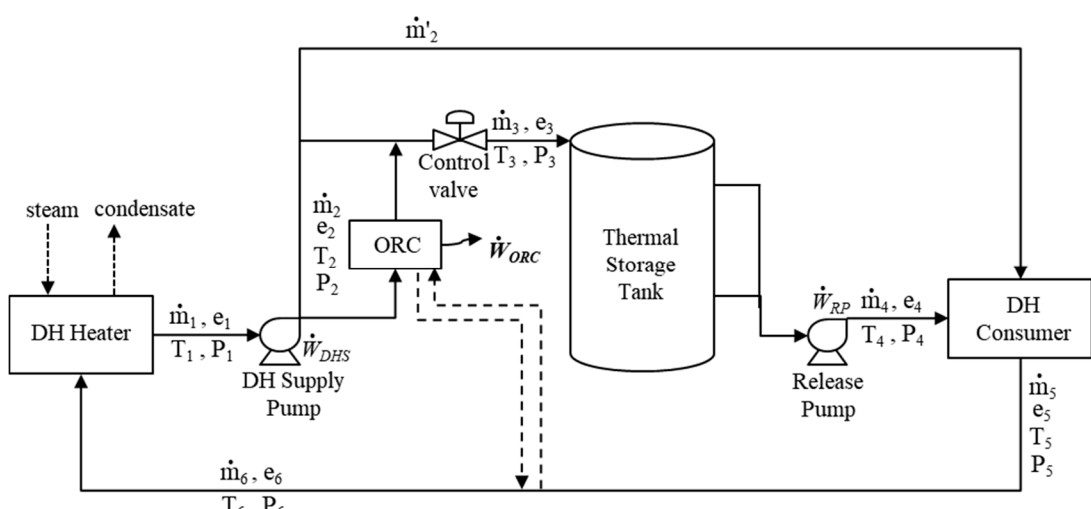

**Figure 5.** Configuration of organic Rankine cycle.

As can be seen in Figure 5, the organic Rankine cycle was applied to Process ②, in which maximum exergy loss in the thermal energy storage process occurred. The organic Rankine cycle uses the primary heating water stored in the TES tank as the heat source. The condenser cooling water is the return DH water from the district heating customer. If the return DH water is used as the cooling water, the condensation heat of the working fluid discharged from the organic Rankine cycle can be recovered and used for preheating the return DH water. Therefore, a more effective system can be configured to reduce the exergy loss of the TES system when the organic Rankine cycle is applied. By the boundary conditions of the organic Rankine cycle, the detailed design conditions were as shown in Table 7.

**Table 7.** Design conditions of organic Rankine cycle.

| Location | Heat Source | | Working Fluid | | Cooling Water | |
|---|---|---|---|---|---|---|
| | Item | Value | Item | Value | Item | Value |
| Inlet | Temperature | 110 °C | Temperature | 69.01 °C | Temperature | 45 °C |
| | Flowrate | 1500 t/h | Pressure | 9.257 ata | Flowrate | 2755 t/h |
| Outlet | Temperature | 100 °C | Temperature | 100 °C | Temperature | 50 °C |
| | Flowrate | 1500 t/h | Pressure | 9.076 ata | Flowrate | 2755 t/h |

When the organic Rankine cycle is applied to the TES system based on the design conditions of Table 7, it is possible to produce 1150.5 kW of electricity from the organic Rankine cycle. In addition, the exergy loss of 3174.4 kW was recovered by increasing the temperature of the primary district heating return water by 5 °C. Thus, the total recovery of exergy loss was 4324.9 kW when using the organic Rankine cycle.

### 3.3. Reduction of Exergy Loss by Pressure Loss

Even if exergy loss due to thermal loss is reduced by applying an organic Rankine cycle, there is still exergy loss due to pressure loss. As previously analyzed, the total exergy loss in Process ② was 4879.6 kW. An organic Rankine cycle was applied to recover the exergy loss of 4324.9 kW but the exergy loss still remained at 554.7 kW. This exergy loss included the pressure loss from the throttling of the valve, the loss of the working fluid expander, internally present in the organic Rankine cycle, and the exergy loss from heat exchange between the working fluid and the cooling water in the condenser. This exergy loss was caused by the exergy loss of the device itself due to the exergy loss from the working fluid expander and the condenser. This is another way to increase the efficiency of the device or to reduce the exergy lost during the heat exchange process. Therefore, it is virtually impossible to recover until the development of more high-efficiency equipment or another way of reducing exergy loss in heat exchange processes. The present study does not discuss the exergy loss reduction problem stemming from the device itself.

Hydraulic turbines have been applied to reduce residual exergy loss. The configuration of a combination of systems using an organic Rankine cycle and hydraulic turbines is shown in Figure 6. The boundary conditions of hydraulic turbines are as follows:

- 　Flow rate of turbine inlet: 1500 m$^3$;
- 　Pressure of turbine inlet: 11.6 kg/cm$^2$g;
- 　Temperature of turbine inlet: 100 °C;
- 　Pressure of turbine outlet: 4.5 kg/cm$^2$g.

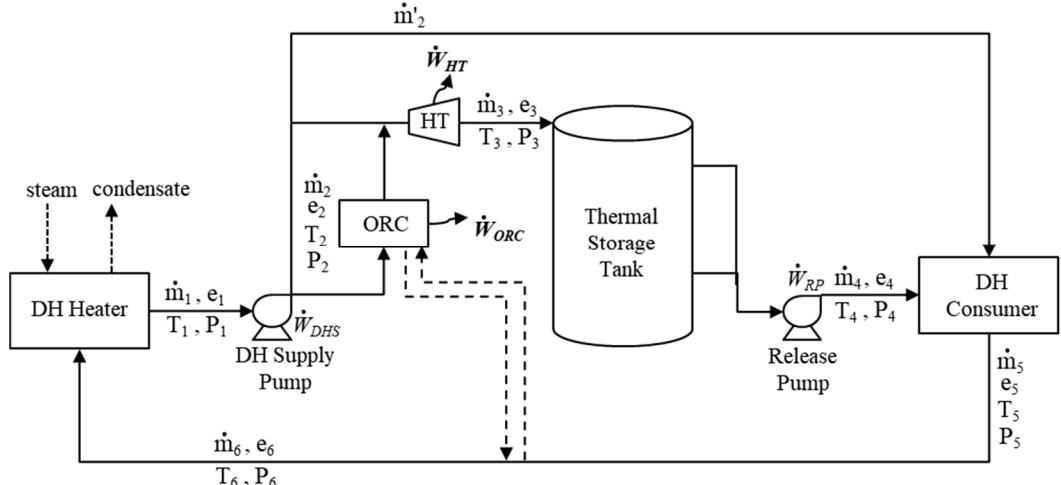

**Figure 6.** Configuration of combination system of organic Rankine cycle and hydraulic turbine.

According to the results of the analysis of the heating water supply pressure in Table 3, the inlet pressure of the hydraulic turbine was applied to the average supply pressure and the turbine outlet pressure was based on Table 5. From the analysis results, it was estimated that a power generation of 208.4 kW was possible with the hydraulic turbine. Hydraulic turbines can be a good alternative to pressure control by replacing the control valve with the reduction of the pressure lost by throttling in the heat storage system.

## 3.4. Economic Evaluation

For the above measures, proposed to reduce the exergy loss in the TES system, it was necessary to carry out an economic analysis to evaluate their applicability. Therefore, an economic analysis of the ORC and a hydraulic turbine was performed in this study.

The payback period method was used for the economic analysis. The payback period (PB) is defined as the time taken for the initial investment cost and product sales revenue to be equal (electric sales revenue in this study). This is defined by Equation (8). Here, $C_0$ denotes the initial investment cost and $i$ denotes the discount rate [24,25]. In this study, $C_0$ was obtained from the manufacturer's proposal and $i$ was assumed to be 5%.

$$PB = \log\left(\frac{GM}{GM - i \times C_0}\right) / \log(1 + i), \tag{8}$$

GM (Gross Margin) means the total revenue of the facility for a year and is defined by the following equation:

$$GM = R_{el} - C_{OM} = (PW \times PR) - C_{OM}, \tag{9}$$

where PW (kWh) is the magnitude of electricity produced by each facility, PR ($/kWh) is the unit price of electricity and $C_{OM}$ ($) denotes the relative operation and maintenance cost.

The electricity sales unit price applied for the economic analysis was based on the Korea Power Exchange's average electricity trading price in 2018 [26]. The maintenance cost was 3% of the facility investment cost.

It was considered appropriate to apply the capacity factor of the combined heat and power (CHP), a heat source facility, but it was necessary to make assumptions about the capacity factor because the acquisition of the capacity factor information for the heat source facility was limited. Thus, the capacity factor of the facility was assumed to be 80% in this study (Table 8).

**Table 8.** Parameters for economic analysis.

| Item | Value |
|---|---|
| Combined heat and power (CHP) Plant Capacity factor | 0.8 |
| Discount rate (%) | 5 |
| Unit Price of Electricity ($/kWh) | 0.1 |
| Operating and Maintenance Cost | 3% of each facility |

The results of economic analysis are shown in Table 9.

**Table 9.** Results of economic analysis.

| Item | ORC | Hydraulic Turbine |
|---|---|---|
| Investment Cost ($) | 1,870,000 | 945,000 |
| Income of Electricity sales ($/yr) | 651,744 | 189,216 |
| Operating and Maintenance Cost ($/yr) | 56,100 | 28,350 |
| Gross Margin ($/yr) | 595,644 | 160,866 |
| Payback Period (yr) | 3.50 | 7.13 |

As a result, the payback period of the ORC and hydraulic turbine will be 3.5 and 7.13 years, respectively, so that both measures can be practically applied as useful option to reduce exergy loss in TES system. However, the hydraulic turbine has a disadvantage in that the facility cost per unit power output and installation cost are high due to the civil engineering work. It is somewhat less economical than the ORC.

This study calculated the theoretical electricity production and economy based on the operating data of temperature and pressure. Although the results are economical, the actual case can affect the electricity production and economy as the site, space, and types of ORC and hydraulic turbine are selected. Therefore, it is necessary to judge the electricity production and economic efficiency through practical design or field demonstration.

## 4. Conclusions

The thermal energy storage (TES) system stores the district heating (DH) water which is primary heating water generated from a combined heat and power (CHP) plant when the thermal load is low. It then releases the district heating water when the thermal load is high. It is a system installed for the efficient operation of heat source facilities. In a general TES system, the high temperature water, 98 °C, and the low temperature water, 65 °C, are stored in the same tank. When heat is being stored, the low temperature water stored in the TES system is sent to the CHP plant (heat source facilities) and then to heated up to the DH supply temperature of 115 °C.

Since a TES system stores heat at atmospheric pressure, the primary heating water of 115 °C produced from a CHP plant has to be lowered to less than 100 °C. Therefore, the temperature of the primary heating water results in heat loss during storage. In addition, the primary heating water must have high pressure to supply heat to DH users a long distance from the CHP plant. When heat must be stored in the TES system a pressure drop in a throttling valve occurs. These losses occur in the thermal storage process of the TES system and can be analyzed by exergy analysis method to identify the locations, causes and the amount of loss. Exergy deals with the qualitative aspects of energy based on the second law of thermodynamics. Exergy analysis can provide a multitude of information to improve the efficiency of the process, identify problems and suggest a direction for improvement.

This study has evaluated the method on the efficiency improvement of a TES system using exergy analysis method in heat storage process. A power generation technology is also proposed using ORC and hydraulic turbine as a way to recover the exergy loss of the TES system. The exergy loss point and loss amount were identified through the point of view of second law of thermodynamics, and the exergy efficiency of the heat storage system was deep-dive analyzed when applying the power generation technology. The appropriate scale of ORC and hydraulic turbine for the heat storage system was suggested.

As a result, the ORC of 930 kW capacity and the hydraulic turbine of 270 kW capacity are considered to be suitable for the heat storage system that stores 3000 m$^3$/h. In this case, each power generation facility is 50% of the thermal storage capacity, which is attributed to the variation of heat storage from the annual operating pattern analysis. Therefore, it is possible to produce 1200 kW of power by recovering 4533.3 kW exergy losses. In addition, economic analysis established that both facilities can reduce exergy losses in the TES system and be practically applicable.

**Author Contributions:** Data Curation, J.Y.Y.; Conceptualization, K.M.K.; Formal Analysis, J.L.; Project Administration, M.S.O.

**Acknowledgments:** This work was supported by the Energy Demand Management Program of the Korea Institute of Energy Technology Evaluation and Planning (KETEP), which was granted financial resources from the Ministry of Trade, Industry and Energy, Republic of Korea (No. 20172010000190).

**Conflicts of Interest:** The authors declare no conflicts of interest.

## Nomenclature

| | |
|---|---|
| $C_{OM}$ | Operation and maintenance cost, $ |
| *DH* | District heating |
| *DHS* | District heating supply |
| e | Exergy per unit mass, kJ/kg |
| $E_{ch}$ | Chemical exergy resulting from the reaction, kJ/kg |
| $E_D$ | Exergy destruction, kJ/kg |
| $E_e$ | Outlet of exergy, kJ/kg |
| $E_i$ | Inlet of exergy, total exergy in the system, kJ/kg |
| $E_{kn}$ | Exergy by velocity or kinetic energy, kJ/kg |
| $E_L$ | Exergy loss, kJ/kg |
| $E_{ph}$ | Physical exergy, difference between actual state and reference state, kJ/kg |
| $E_{pt}$ | Exergy by the potential energy, kJ/kg |
| GM | Gross margin, $/yr |
| *HT* | Hydraulic turbine |
| *i* | Additional ratio, % |
| $\dot{m}$ | Flow rate, kg/s |
| P | Pressure, kg/cm²g |
| PB | Payback period, $/kWh |
| PW | Magnitude of electricity, kWh |
| *RP* | Release pump |
| Q | Heat transfer, kJ |
| T | Temperature, °C |

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
