# Peer review of "Exergy Analysis for Utilizing Latent Energy of Thermal Energy Storage System in District Heating"

_energies, doi:10.3390/en12071391_

Round 1
Reviewer 1 Report
The topic is interesting, however there are a few points that Authors should address.
Abstract should be rewritten with the first sentence providing significance of the work or a broad background.
Introduction should be restructured to define relevance, impact and significance of the work. The flow of information in introduction is not appropriate.
Figures have a caption, but have not explanation of markings.
Conclusions should be supported by experimental verification of the results obtained from the analysis performed, thus providing goodness of the proposals developed.
How the results of exergy analysis can be validated in an industrial application?
Author Response
Answers to the Reviewer 1’s Comments
Title: Exergy Analysis for Utilizing Latent Energy of Thermal Energy Storage System in District Heating
Number: energies-471441
Authors: Joong Yong Yi, Kyung Min Kim, Jongjun Lee, Mun Sei OH
We appreciate the significant efforts that the reviewers put into our paper to review. We made the changes suggested by the reviewers to improve the quality of our paper.
(Comment 1)
Abstract should be rewritten with the first sentence providing significance of the work or a broad background.
(Answer)
We revised the abstract as follows:
“The thermal energy storage (TES) system stores the district heating (DH) water when the heating load is low. Since a TES system stores heat at atmospheric pressure, the DH water of 115°C has to be changed into the water less than 100°C. Therefore, the temperature drop of the DH water results in thermal loss during storage. In addition, the DH water must have high pressure to supply heat to DH users with long distance from the CHP plant. If heat is to be stored in the TES system, the pressure drop in a throttling valve occurs in this process. These exergy losses, which occur in the thermal storage process of the general TES system, can be analyzed by exergy analysis to identify the location, causes and the amount of losses. This study evaluated the method on the efficiency improvement of a TES system through exergy calculation in heat storage process. The method is that a power generation technology using organic Rankine cycle (ORC) and hydraulic turbine. As a result, the ORC of 930kW capacity and the hydraulic turbine of 270kW capacity are considered to be suitable for the heat storage system that stores 3,000m3/hr. In this case, each power generation facility is 50% of the thermal storage capacity, which is attributed to the variation of actual heat storage from the annual operating pattern analysis. Therefore, it is possible to produce 1,200 kW of power by recovering the exergy losses. The payback period of ORC and hydraulic turbine is 3.5 and 7.13 years, respectively.”
(Comment 2)
Introduction should be restructured to define relevance, impact and significance of the work. The flow of information in introduction is not appropriate.
(Answer)
We revised the paragraph as follow:
“The previous studies have studied new design of TES system, new materials, and solar thermal storage system. In present, however, many TES systems for storing hot water have been used at atmospheric pressure in industrial fields. This study has investigated the energy loss generated in the process of storing heat in DH industries. A method to improve the efficiency of the TES system by utilizing the potential energy have been studied. Using exergy analysis method, the exergy values of the respective streams constituting the target system are calculated. The points where the exergy losses occurred and the amount of losses were also calculated in the TES system. In addition, the methods of reducing exergy losses in each process is proposed and additional generated energies are calculated.”
(Comment 3)
Figures have a caption, but have not explanation of markings.
(Answer)
We added “Nomenclature.”
(Comment 4)
Conclusions should be supported by experimental verification of the results obtained from the analysis performed, thus providing goodness of the proposals developed.
(Answer)
We added the limitation (experimental verification) of the study in the sections of “Results and Discussion” and “Conclusions”.
“This study calculated the theoretical electricity production and economy based on operating data of temperature and pressure. Although the results are economical, the actual case can affect the electricity production and economy as the site, space, and types of ORC and hydraulic turbine are selected. Therefore, it is necessary to judge the electricity production and economical efficiency through practical design or field demonstration.”
“This study has evaluated the method on the efficiency improvement of a TES system using exergy analysis method in heat storage process. A power generation technology is also proposed using ORC and hydraulic turbine as a way to recover the exergy loss of the TES system. The exergy loss point and loss amount were identified through the point of view of second law of thermodynamics, and the exergy efficiency of the heat storage system was deep-dive analyzed when applying the power generation technology. The appropriate scale of ORC and hydraulic turbine for the heat storage system was suggested.
As a result, the ORC of 930kW capacity and the hydraulic turbine of 270kW capacity are considered to be suitable for the heat storage system that stores 3,000m3/hr. In this case, each power generation facility is 50% of the thermal storage capacity, which is attributed to the variation of heat storage from the annual operating pattern analysis. Therefore, it is possible to produce 1,200kW of power by recovering 4,533.3 kW exergy losses. In addition, it was found from economic analysis that both facilities can be good measures to reduce exergy losses in TES system and be practically applicable.”
(Comment 5)
How the results of exergy analysis can be validated in an industrial application?
(Answer)
We estimated the results of exergy analysis from numerical simulation. This simulation used the actual data, such as actual operating data (temperature, pressure and flowrates) of thermal energy storage system and efficiency of ORC and hydraulic turbine systems with similar capacity. However, the actual values may vary depending on the piping configuration and insulation condition of the site, but they are considered to be almost similar to industrial application.

Reviewer 2 Report
1. Lines 13 - 19
“Since a TES system stores heat at atmospheric pressure, the primary heating water of 115°C produced from a CHP plant has to be changed the water less than 100°C. Therefore, the temperature of the primary heating water results in thermal loss during storage. In addition, the primary heating water must have high pressure to supply heat to DH users with long distance from the CHP plant. If heat is to be stored in the TES system, the pressure drop (loss) in a throttling valve occurs in this process. These exergy losses, which occur in the thermal storage process of the TES system, can be analyzed by exergy analysis to identify the location, causes and the amount of losses.”
Please reduce the theoretical part of the abstract. Focus more on your study. Please include more numerical results in the latter part of the abstract.
2. Lines 48 -51
Phase-change materials (PCMs) in solar system
3. Lines 96 – 103. Other studies that have been performed that use advanced thermodynamic analysis for thermal storage. Examples:
Huiru Wang, Zhenyu Liu, Huiying Wu, Entransy dissipation-based thermal resistance optimization of slab LHTES system with multiple PCMs arranged in a 2D array, Energy, Volume 138, 2017, Pages 739-751
Bin Li, Xiaoqiang Zhai, Xiwen Cheng, Thermal performance analysis and optimization of multiple stage latent heat storage unit based on entransy theory, International Journal of Heat and Mass Transfer, Volume 135, 2019, Pages 149-157
4. Lines 96 – 120, please adjust the paragraphs for consistent size
5. Highlight the research gap and the novelty of the study
6. Rename sections “3. Exergy Analysis of TES” and “4. Analysis for reduction of exergy loss in TES Systems” as “3.Results” and “4.Discussion”. Please adjust the content of the sections accorsdingly. The structure of the manuscript should be standardized.
7. In the latter part of section 4 please describe the limitations of the study and policy implications and future work.
8. Please abbreviate the journal names in the reference list.
Author Response
Answers to the Reviewer 2’s Comments
Title: Exergy Analysis for Utilizing Latent Energy of Thermal Energy Storage System in District Heating
Number: energies-471441
Authors: Joong Yong Yi, Kyung Min Kim, Jongjun Lee, Mun Sei OH
We appreciate the significant efforts that the reviewers put into our paper to review. We made the changes suggested by the reviewers to improve the quality of our paper.
(Comment 1) Lines 13 - 19
“Since a TES system stores heat at atmospheric pressure, the primary heating water of 115°C produced from a CHP plant has to be changed the water less than 100°C. Therefore, the temperature of the primary heating water results in thermal loss during storage. In addition, the primary heating water must have high pressure to supply heat to DH users with long distance from the CHP plant. If heat is to be stored in the TES system, the pressure drop (loss) in a throttling valve occurs in this process. These exergy losses, which occur in the thermal storage process of the TES system, can be analyzed by exergy analysis to identify the location, causes and the amount of losses.”
Please reduce the theoretical part of the abstract. Focus more on your study. Please include more numerical results in the latter part of the abstract.
(Answer)
We revised the abstract. We reduced the theoretical part and added numerical results as follows:
“The thermal energy storage (TES) system stores the district heating (DH) water when the heating load is low. Since a TES system stores heat at atmospheric pressure, the DH water of 115°C has to be changed into the water less than 100°C. Therefore, the temperature drop of the DH water results in thermal loss during storage. In addition, the DH water must have high pressure to supply heat to DH users with long distance from the CHP plant. If heat is to be stored in the TES system, the pressure drop in a throttling valve occurs in this process. These exergy losses, which occur in the thermal storage process of the general TES system, can be analyzed by exergy analysis to identify the location, causes and the amount of losses. This study evaluated the method on the efficiency improvement of a TES system through exergy calculation in heat storage process. The method is that a power generation technology using organic Rankine cycle (ORC) and hydraulic turbine. As a result, the ORC of 930kW capacity and the hydraulic turbine of 270kW capacity are considered to be suitable for the heat storage system that stores 3,000m3/hr. In this case, each power generation facility is 50% of the thermal storage capacity, which is attributed to the variation of actual heat storage from the annual operating pattern analysis. Therefore, it is possible to produce 1,200 kW of power by recovering the exergy losses. The payback period of ORC and hydraulic turbine is 3.5 and 7.13 years, respectively.”
(Comment 2)
Lines 48 -51: Phase-change materials (PCMs) in solar system
Lines 96 – 103. Other studies that have been performed that use advanced thermodynamic analysis for thermal storage.
(Answer)
We added the references as reviewer mentioned.
“Other studies [7, 8] that have been performed that use advanced thermodynamic analysis for thermal storage. They found the optimal values of PCMs.”
(Comment 3)
Lines 96 – 120, please adjust the paragraphs for consistent size
(Answer)
We adjusted the paragraphs for consistent size.
(Comment 4)
Highlight the research gap and the novelty of the study
(Answer)
We revised the paragraph as follow:
“The previous studies have studied new design of TES system, new materials, and solar thermal storage system. In present, however, many TES systems for storing hot water have been used at atmospheric pressure in industrial fields. This study has investigated the energy loss generated in the process of storing heat in DH industries. A method to improve the efficiency of the TES system by utilizing the potential energy have been studied. Using exergy analysis method, the exergy values of the respective streams constituting the target system are calculated. The points where the exergy losses occurred and the amount of losses were also calculated in the TES system. In addition, the methods of reducing exergy losses in each process is proposed and additional generated energies are calculated.”
(Comment 5)
Rename sections “3. Exergy Analysis of TES” and “4. Analysis for reduction of exergy loss in TES Systems” as “3.Results” and “4.Discussion”. Please adjust the content of the sections accorsdingly. The structure of the manuscript should be standardized.
(Answer)
We revised the section names, “2. Research methods”, “3. Results and Discussion”, etc.
(Comment 6)
In the latter part of section 4 please describe the limitations of the study and policy implications and future work.
(Answer)
We added the limitation of the study and future work.
“This study calculated the theoretical electricity production and economy based on operating data of temperature and pressure. Although the results are economical, the actual case can affect the electricity production and economy as the site, space, and types of ORC and hydraulic turbine are selected. Therefore, it is necessary to judge the electricity production and economical efficiency through practical design or field demonstration.”
(Comment 7)
Please abbreviate the journal names in the reference list.
(Answer)
We revised the reference list.

Reviewer 3 Report
The authors have identified loss of exergy in the system shown in Figure 2.
The authors have determined the losses of exergy and then, in order to limit them, they proposed a system expanded with an ORC cycle and a hydraulic turbine.
Of course, the losses of exergy should be reduced, but within the limits of economic viability and technological possibilities.
After reading the article in its current form, I'm not satisfied. I have more questions than answers.
The paper lacks cost calculations.
Is it profitable to add ORC and hydraulic turbine, what will be the cost of construction, what will be the profit?
I
suggest expanding the article. Calculate the cost of building a basic
and extended variant. It should be proofed that it is profitable during
the operation of the system.
This will give an interesting effect in the form of an answer to the question whether it is worth expanding the system.?
English:
Please avoid starting sentences with "We". The passive voice should be used.
Table 4 is unreadable.
Author Response
Answers to the Reviewer 3’s Comments
Title: Exergy Analysis for Utilizing Latent Energy of Thermal Energy Storage System in District Heating
Number: energies-471441
Authors: Joong Yong Yi, Kyung Min Kim, Jongjun Lee, Mun Sei OH
We appreciate the significant efforts that the reviewers put into our paper to review. We made the changes suggested by the reviewers to improve the quality of our paper.
(Comment 1)
The authors have identified loss of exergy in the system shown in Figure 2. The authors have determined the losses of exergy and then, in order to limit them, they proposed a system expanded with an ORC cycle and a hydraulic turbine. Of course, the losses of exergy should be reduced, but within the limits of economic viability and technological possibilities. After reading the article in its current form, I'm not satisfied. I have more questions than answers.
The paper lacks cost calculations. Is it profitable to add ORC and hydraulic turbine, what will be the cost of construction, what will be the profit? I suggest expanding the article. Calculate the cost of building a basic and extended variant. It should be proofed that it is profitable during the operation of the system.
(Answer)
We added the section of “3.4 Economic evaluation.”
(Comment 2)
This will give an interesting effect in the form of an answer to the question whether it is worth expanding the system.?
(Answer)
This study has evaluated the method on the efficiency improvement of a TES system using exergy analysis method in heat storage process. A power generation technology is also proposed using ORC and hydraulic turbine as a way to recover the exergy loss of the TES system. The exergy loss point and loss amount were identified through the point of view of second law of thermodynamics, and the exergy efficiency of the heat storage system was deep-dive analyzed when applying the power generation technology. The appropriate scale of ORC and hydraulic turbine for the heat storage system was suggested.
As a result, the ORC of 930kW capacity and the hydraulic turbine of 270kW capacity are considered to be suitable for the heat storage system that stores 3,000m3/hr. In this case, each power generation facility is 50% of the thermal storage capacity, which is attributed to the variation of heat storage from the annual operating pattern analysis. Therefore, it is possible to produce 1,200kW of power by recovering 4,533.3 kW exergy losses. In addition, it was found from economic analysis that both facilities can be good measures to reduce exergy losses in TES system and be practically applicable.
(Comment 3)
English: Please avoid starting sentences with "We". The passive voice should be used.
(Answer)
We changed the sentences as follows:
Line 17, “This study evaluated the method on the efficiency improvement of a TES system through exergy calculation in heat storage process.”
Line 83, “there are two energy losses, such as heat loss and pressure loss in the heat-storage process of a district heating system.”
Line 93, “It is also able to get a lot of information and obtain an optimum design for efficiency improvement of the TES system.”
Line 94, “A number of studies related to district heating systems have been undertaken.”
Line 112, “Firstly, it is required to analyze the energy losses in temperature, pressure, electricity, or any forms in a system, and then find ways to improve efficiency through energy recovery.”
Line 117, “A method to improve the efficiency of the TES system by utilizing the potential energy have been studied.”
Line 210, “the flow rate of the heating water from the amount of storing heat can be calculated
(Comment 4)
Table 4 is unreadable.
(Answer)
The wrong data in Table 4 was corrected.

Round 2
Reviewer 1 Report
They Authors made superficial changes in the manuscript.
Reviewer 3 Report
I Accept article in present form.